# Optimizing tuberculosis treatment efficacy: Comparing the standard regimen with Moxifloxacin-containing regimens

**Maral Budak**[1], **Joseph M. Cicchese**[2], **Pauline Maiello**[3], **H. Jacob Borish**[3], **Alexander G. White**[3], **Harris B. Chishti**[3], **Jaime Tomko**[3], **L. James Frye**[3], **Daniel Fillmore**[3], **Kara Kracinovsky**[3], **Jennifer Sakal**[3], **Charles A. Scanga**[3], **Philana Ling Lin**[3], **Véronique Dartois**[4,5], **Jennifer J. Linderman**[2], **JoAnne L. Flynn**[3], **Denise E. Kirschner**[1]*

1 Department of Microbiology and Immunology, University of Michigan Medical School, Ann Arbor, Michigan, United States of America, 2 Department of Chemical Engineering, University of Michigan, Ann Arbor, Michigan, United States of America, 3 Department of Microbiology and Molecular Genetics and Center for Vaccine Research, University of Pittsburgh School of Medicine, Pittsburgh, Pennsylvania, United States of America, 4 Center for Discovery and Innovation, Hackensack Meridian Health, Nutley, New Jersey, United States of America, 5 Department of Medical Sciences, Hackensack Meridian School of Medicine, Nutley, New Jersey, United States of America

* kirschne@umich.edu

**Data Availability Statement:** All data are in the manuscript and/or supporting information files.

**Funding:** This research was supported by NIH Grants R01 AI50684 (DEK, JLF), U01 HL131072

## Abstract

Tuberculosis (TB) continues to be one of the deadliest infectious diseases in the world, causing ~1.5 million deaths every year. The World Health Organization initiated an *End TB Strategy* that aims to reduce TB-related deaths in 2035 by 95%. Recent research goals have focused on discovering more effective and more patient-friendly antibiotic drug regimens to increase patient compliance and decrease emergence of resistant TB. Moxifloxacin is one promising antibiotic that may improve the current standard regimen by shortening treatment time. Clinical trials and *in vivo* mouse studies suggest that regimens containing moxifloxacin have better bactericidal activity. However, testing every possible combination regimen with moxifloxacin either *in vivo* or clinically is not feasible due to experimental and clinical limitations. To identify better regimens more systematically, we simulated pharmacokinetics/pharmacodynamics of various regimens (with and without moxifloxacin) to evaluate efficacies, and then compared our predictions to both clinical trials and nonhuman primate studies performed herein. We used *GranSim*, our well-established hybrid agent-based model that simulates granuloma formation and antibiotic treatment, for this task. In addition, we established a multiple-objective optimization pipeline using *GranSim* to discover optimized regimens based on treatment objectives of interest, i.e., minimizing total drug dosage and lowering time needed to sterilize granulomas. Our approach can efficiently test many regimens and successfully identify optimal regimens to inform pre-clinical studies or clinical trials and ultimately accelerate the TB regimen discovery process.

(DEK, VAD, JJL, JLF), S10 OD023524 (VAD) and supported in part by funding by the Wellcome Leap ΔTissue Program awarded to (DEK, JJL, JLF). Further funding comes from a contract grant from the MRI Bill & Melinda Gates Foundation awarded to DEK. The funders had no role in study design, data collection and analysis, decision to publish, or preparation of the manuscript.

**Competing interests:** None.

## Author summary

Tuberculosis (TB) is a top global health concern and the WHO has made END TB a goal for 2050. Treatment for TB requires multiple antibiotics taken for long periods of time, which is challenging for TB patients due to side effects and compliance issues. Therefore, identifying regimens that are more effective and more patient-friendly than the currently used 4-drug standard regimen treatment is urgently needed. It is also known that non-compliance leads to the development of drug resistant TB. In this work, we first apply our next-generation computational model that captures the immune response to infection in lungs with *M. tuberculosis* via the formation of granulomas to predict new regimens for the treatment of TB. These include regimens that have been recently tried in clinical trials with controversial results. Our goal is to identify regimens that optimize how fast bacteria are cleared using minimal dosages. We then pair our studies with the best experimental system for TB, namely, validating our predictions using a non-human primate model. Our findings suggest new regimens and additionally that systems pharmacological modeling should be employed as a method to narrow the design space for drug regimens for tuberculosis and other diseases as well prior to clinical trials.

## Introduction

Tuberculosis (TB) is one of the deadliest infectious diseases in the world, with 1.6 million deaths in 2021 [1], and World Health Organization (WHO) aims to reduce the number of TB-related deaths by 95% by 2035 [1]. While vaccination efforts can reduce the number of new TB cases and deaths, a shorter but highly efficacious and safe drug regimen is needed to treat TB. Although new and efficacious drugs have been discovered for drug-resistant TB [2,3], drug-susceptible TB disease has been treated with the same regimen for close to 50 years, namely 6–9 months of treatment with isoniazid (H), rifampin (R), ethambutol (E) and pyrazinamide (Z) [4]. Likely, changes to the existing standard regimen for drug-susceptible TB will help achieve WHO's goal.

Improving existing TB treatment involves finding regimens that account for the complexities of TB. The structure of the granuloma influences antibiotic distribution and can result in lower concentrations within granulomas [5–8]. Moreover, microenvironments within granulomas can promote the infecting bacteria, *Mycobacterium tuberculosis* (Mtb), to shift phenotypic states that are tolerant towards antibiotics [9–11]. Host-to-host variability in drug absorption and metabolism kinetics leads to pharmacokinetic (PK) variability that has been clinically linked to worse outcomes in TB treatment [12]. Furthermore, the lengthy treatment makes compliance challenging. While compliance yields high levels of success, intermittent treatment can lead to the development of drug resistance [13]. In short, by addressing these complications (heterogeneity in granulomas and antibiotic distribution, antibiotic-tolerant Mtb, host-to-host PK variability and long treatment times), a better regimen–one that would successfully treat more individuals with a shorter treatment duration–can be identified.

Due to these challenges, identifying new regimens for TB is a complex process that requires a combination of approaches to accurately capture different aspects of TB treatment [14]. Studies have classified the pharmacokinetic/ pharmacodynamic (PK/PD) features of individual TB antibiotics with *in vitro* methods, such as hollow fiber systems [15–17] and bactericidal assays in different growth conditions [18–20], and *in vivo* methods via HPLC coupled to tandem mass spectrometry (LC-MS/MS) and MALDI mass spectrometry imaging (MALDI-MSI) analyses [21–23]. However, these studies were mostly performed using single antibiotics and,

due to heterogeneity among granulomas, variability of Mtb metabolic states and the propensity for Mtb to develop drug resistance, TB treatments with more than one antibiotic (i.e. combination therapy) are essential. To quantify drug interactions and assess the efficacy of combination therapies, many studies have been performed: *in vitro* with checkerboard assays [24–26], *in vivo* with mouse [27,28], using a non-human primate (NHP) animal models [29,30], as well as *in silico* approaches applying machine learning algorithms [31,32]. Moreover, many clinical studies have been performed with antibiotic combinations, which is crucial to assessing toxicity as well as long term outcomes of treatments [33–36]. These valuable studies are time-consuming and expensive, often prohibitively so.

Computational modeling can efficiently predict regimen efficacy and optimal doses, which is essential due to the high number of combinations of drug regimens in this large regimen design space (on the order of $10^{17}$ [37]). We have previously shown that our validated computational simulations of granuloma formation, function and treatment, called *GranSim*, can simulate efficacies of different TB regimens (c.f. [6,8,38]) and we can utilize surrogate-assisted optimization algorithms to accurately and efficiently predict optimal regimens [37].

Previous studies in murine models suggested that moxifloxacin (M) is a promising antibiotic to improve the standard regimen and decrease the duration of TB treatment due to its strong bactericidal activity [39–44]. To this end, a recent clinical trial, REMoxTB, attempted to shorten treatment from 6 months to 4 months by altering the standard HRZE regimen to HRZM or RMZE. However, the study failed to show noninferiority of moxifloxacin-containing regimens to the standard regimen due to higher relapse rates of these regimens [33]; after careful reanalysis, some patient populations were shown to be cured successfully with these moxifloxacin-containing regimens in a shorter treatment window [45]. In our study, we elaborate an approach toward identifying drug regimens that are more effective in treating TB granulomas and that require shorter treatment times compared to the standard regimen. We used our computational model *GranSim* to create an *in silico* biorepository of hundreds of granulomas, combined with *in vivo* data generated from a NHP model and applied a surrogate-assisted optimization algorithm to identify regimen success and failure. We first simulated moxifloxacin-containing regimens using *GranSim* and identified regimens that are superior to the standard treatment based on sterilization times. Informed by our simulation results, we performed an *in vivo* study in NHPs to test our predicted regimens that haven't been studied before, validating our simulation predictions. Thus, our study identifies new regimens that can inform pre-clinical trials to shorten treatment times and minimize dosages. This highlights the importance of using modeling prior to pre-clinical trials as a step towards a more efficient and directed regimen design for TB.

## Results

### *In silico* library of granulomas for treatment simulations and dose optimization

We first generated an *in silico* library of 750 granulomas over 300 days that matches NHP dataset of 600 granulomas [46,47]. To do that, we sampled 250 granuloma parameter sets within biological feasible ranges using the LHS method. We varied parameters that determine a variety of host immune responses during Mtb infection (e.g., immune cell recruitment, macrophage apoptosis rate etc.) and metabolic activity (e.g., chemokine/cytokine production rate) (see Table 1 in [8] for the full list of parameters that are varied). Then, we simulated three replications with each parameter set to capture both types of uncertainty present [48]. As a result of simulating *GranSim* with these parameter sets, granulomas emerge with variable dynamics of CFU/immune cell counts and granuloma sizes. We then classified granulomas that have

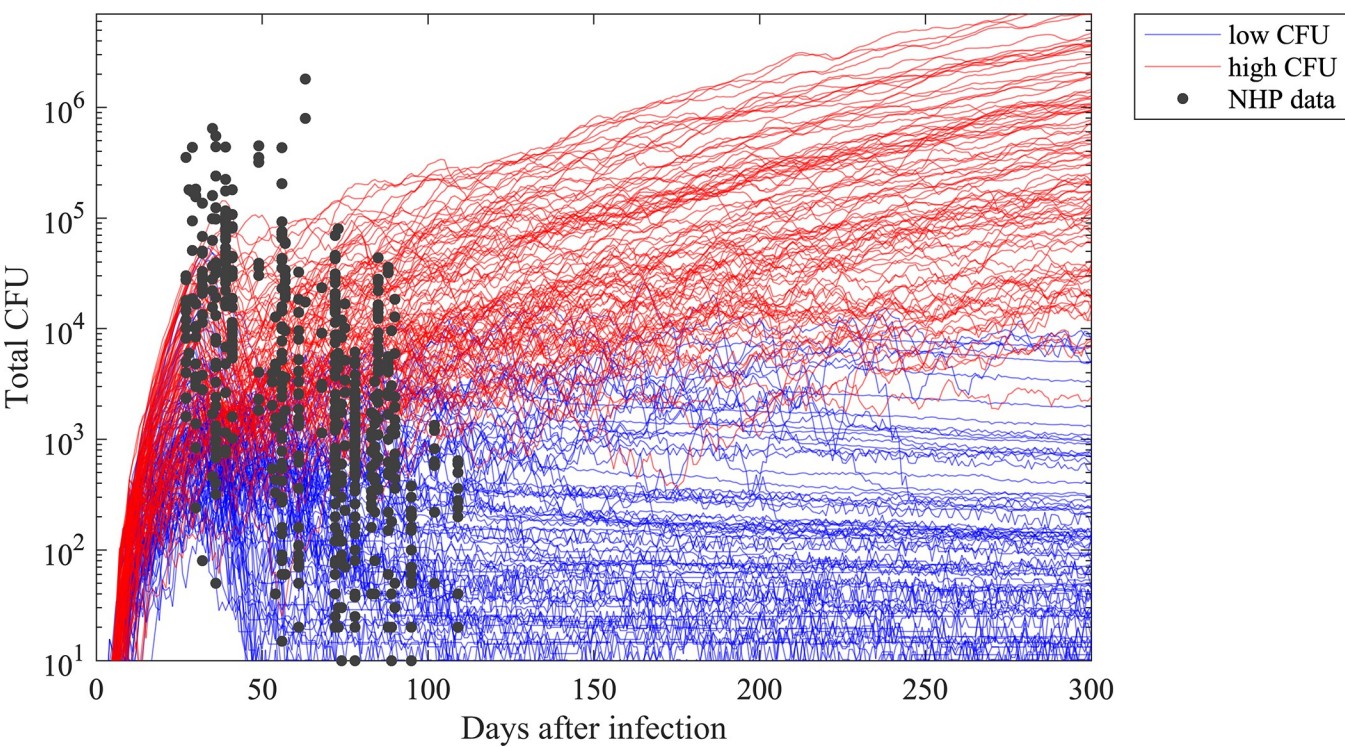

**Fig 1. CFU trends within the *in silico* repository of simulated granuloma generated by *GranSim* after the start of infection.** Each curve represents a single granuloma simulation with a single parameter set using *GranSim*, and black dots are CFU counts from NHP granulomas [46,47]. Each individual data point comes from a granuloma of a unique NHP; however, multiple data points at the same time step may come from the same NHP if that NHP has developed multiple granulomas by the time they were necropsied. In total, the data points come from 42 monkeys and 646 granulomas, and each monkey has 2–40 granulomas (the median is 14.5, 25th and 75th percentiles are 9 and 20, respectively.). Based on their CFU trajectories, we categorize granulomas into low–CFU (blue curves, N = 100) and high–CFU (red curves, N = 100) granulomas. Low–CFU granulomas represent granulomas that have controlled bacterial burden; high–CFU granulomas are those where bacterial growth is uncontrollable by the immune system, respectively [8,49,50].

nonzero bacterial loads (those that did not sterilize) by measuring their colony forming units (CFUs) as either *low-CFU* or *high-CFU granulomas*, depending on their CFU trends (Fig 1) In this work, we simulated different treatments on subsets of granulomas from this library of both high- and low-CFU granulomas as well as combined. This follows as humans and NHPs have multiple granulomas within their lungs, and ensures that we test each regimen on a variety of granuloma types and multiple granulomas, making it relevant to both experimental data and clinical TB outcomes. Here, low-CFU granulomas represent the state where the immune system controls bacterial growth within a granuloma, whereas within high-CFU granulomas, bacteria grow to large numbers and can disseminate [8, 49, 50]. Specifically, if the number of CFUs within a granuloma is less than $10^4$ at the end of the simulation and has not increased more than 50 CFUs in the last 20 days of simulation, we label it as a *low-CFU granuloma* (Fig 1, blue curves). If the number of CFUs in a granuloma is between $10^4$ and $10^7$ at the end of the simulation or it has increased by more than 50 CFUs in the last 20 days of simulation, we label it as a *high-CFU granuloma (*Fig 1, red curves). We proposed $10^4$ CFUs/granuloma as a threshold for low-CFU granulomas, based on the observed CFU trends of the 750 granulomas we simulated: granulomas with CFUs lower than this threshold tend to stabilize in our simulations (Fig 1, blue curves), representing controlled growth. However, granulomas with CFUs higher than this threshold tend to grow uncontrollably (Fig 1, red curves). We can alter this threshold without loss of generality. We randomly selected 100 low-CFU and 100 high-CFU granulomas to simulate regimen treatments.

## Simulations capture the rapid rate of sterilization with moxifloxacin-containing regimens that is observed in clinical trials

We first compare the standard regimen for TB, i.e., HRZE, with various moxifloxacin-containing regimens. A recent clinical trial (REMoxTB) compared the 6-month standard regimen HRZE treatment (control group) to 4-month treatment with two moxifloxacin-containing regimens, HRZM (termed the "isoniazid group" in the original study) and RMZE (termed the "ethambutol group" in the original study) (see Table 1 for the protocol) [33]. Regimens with moxifloxacin were not found to be suitable replacements for the standard regimen, as they had a higher rate of relapse in patients after the end of treatment, even though they decreased the bacterial load in patients' sputum more rapidly at the beginning of the treatment (Fig 2A).

We used *GranSim* to simulate the same protocol as in the REMoxTB study (see Methods and Table 1). Our results agree with the clinical trial: moxifloxacin-containing regimens reduced the bacterial load faster, as ~40% of the granulomas were sterilized within the first week (Fig 2B, dashed red and dotted green curves). By comparison, the standard regimen required more than 4 weeks to sterilize the same number of granulomas (Fig 2B, blue curve). To treat the whole set of granulomas successfully (i.e., both low-and high-CFU), HRZE, HRZM and RMZE-treated groups need 17, 8 and 12 weeks of treatment, respectively. The metric "time to sterilize a granuloma" follows a similar trend: HRZM-treated group has the shortest sterilization time with ~14 days (median), followed by RMZE- (~17 days median) and HRZE-treated (~35 days median) groups. Therefore, our simulations suggest that the HRZM is the most effective regimen in terms of bactericidal activity, followed by RMZE, although the difference between these two regimens is minimal yet significant ($p < 0.001$). The control group, HRZE, is the slowest to sterilize granulomas.

In the REMoxTB study, noninferiority of moxifloxacin-containing regimens was not shown due to the higher relapse rates [33]. Our granuloma-scale model limits our ability to predict disease relapse because it does not contain the dynamics of Mtb-containing granulomas in lymph nodes, which have been shown to induce reinfection when Mtb is present in lymph node granulomas [51] (see Discussion).

## Regimens HMZE, HRZE and RMZE reduce bacterial burden in both NHP studies and simulations

NHPs with active TB were treated with the TB standard regimen HRZE as well as two moxifloxacin-containing regimens: RMZE and HMZE (see Table 1 and Methods). Daily

**Table 1. Simulation protocols used in this study.** Those indicated as clinical trial correspond to the regimens used in [33], and those indicated as NHP study correspond to the regimens tested in NHPs herein. HRZEM combinations refer solely to the computational studies. Optimization refers to the regimens we further tested with our optimization protocol to determine dosing and sterilization time to predict the best performers.

| STUDY | GROUP | REGIMEN |
|---|---|---|
| Clinical trial | Control (HRZE) | 3 wk HRZE + 18 wk HR |
| | HRZM | 17 wk HRZM + 9 wk placebo |
| | RMZE | 17 wk RMZE + 9 wk placebo |
| NHP study | Control (no drug) | 60 days |
| | HMZE | 60 days |
| | HRZE | 60 days |
| | RMZE | 60 days |
| HRZEM combinations | 4-way combinations | 120 days |
| | 3-way combinations | 220 days |
| | 2-way combinations | 300 days |
| Optimization | 4-way combinations | 180 days |

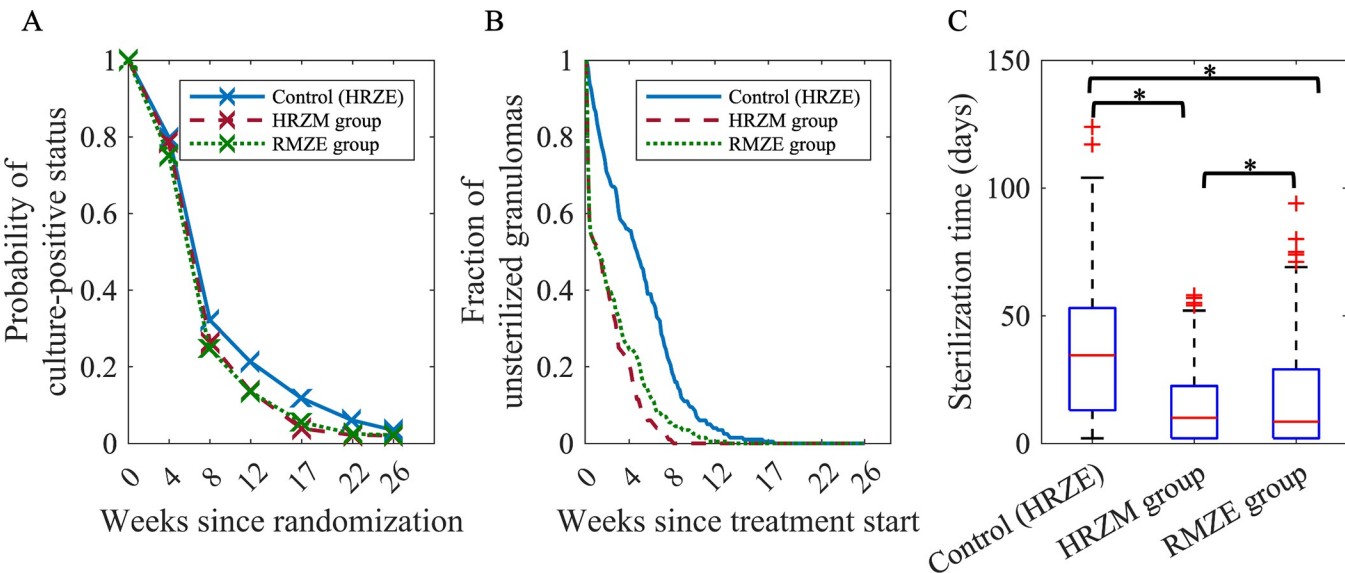

**Fig 2. Comparison of moxifloxacin–containing regimens to the standard regimen for the human study and *GranSim*.** (A) Results from the REMoxTB clinical trial [33]. Probability that a patient has a sputum culture–positive status decreases over the course of treatment, and this decline is more pronounced for moxifloxacin–containing regimens. Control (HRZE), HRZM and RMZE groups have 510, 514 and 524 patients, respectively. This figure is adapted from Fig 2B of [33] (Data points (x) extracted by WebPlotDigitizer). (B,C) *GranSim* predictions for (B) the fraction of unsterilized granulomas and (C) sterilization times upon treatment with HRZE, HRZM and RMZE (*p<0.001, one–tailed paired t–test). The central red lines in box plots represent the median, whereas the bottom and the top edges of boxes represent 25th and 75th percentiles, respectively. For the REMoxTB study and the simulations, in the control groups, patients/granulomas are treated with HRZE for 8 weeks, followed by an 18–week long HR treatment. In HRZM and RMZE groups, patients/granulomas are treated with HRZM and RMZE for 17 weeks, respectively (see Methods and Table 1). In (B) and (C), each group has 200 simulated granulomas.

administration of drugs was initiated at 13 weeks post-infection and continued for 8 weeks at which time the macaques were necropsied. Total CFU was calculated by summing the CFU counts obtained from plating multiple tissue samples (lung, granulomas, LNs) from each animal. Each regimen was able to reduce bacterial burden in NHPs compared to controls (Fig 3D and 3E). Simulations with *GranSim* indicated that moxifloxacin-containing regimens, HMZE and RMZE, sterilize more granulomas in a shorter time frame than the standard regimen, HRZE (Fig 3A–3C).

## Metabolic activity within granulomas is decreased with antibiotic treatment in both NHPs and simulations

We used PET-CT imaging on NHPs with FDG uptake to assess how drug regimens influence inflammatory activity of granulomas. We measured standardized uptake value ratio (SUVR), a previously developed measure to quantify the FDG avidity per granuloma [30,52]. Treatment with HMZE reduced FDG avidity of granulomas within 8 weeks, whereas there was no change in FDG avidity in response to RMZE or HRZE treatment, similar to that of the control group (Fig 4A). In *GranSim*, we monitor metabolic activity of a granuloma based on cells associated with inflammation within granulomas, such as the number of various cell types and inflammatory measures of activity within granulomas (see Methods for more details). Similar to the FDG PET-CT results from NHP experiments, *GranSim* simulations demonstrated that treatment with HMZE decreases metabolic activity significantly (Fig 4B). However, *GranSim* predicted reduced inflammation with RMZE and HRZE as well, unlike NHP experiments, where no change was observed compared to the control.

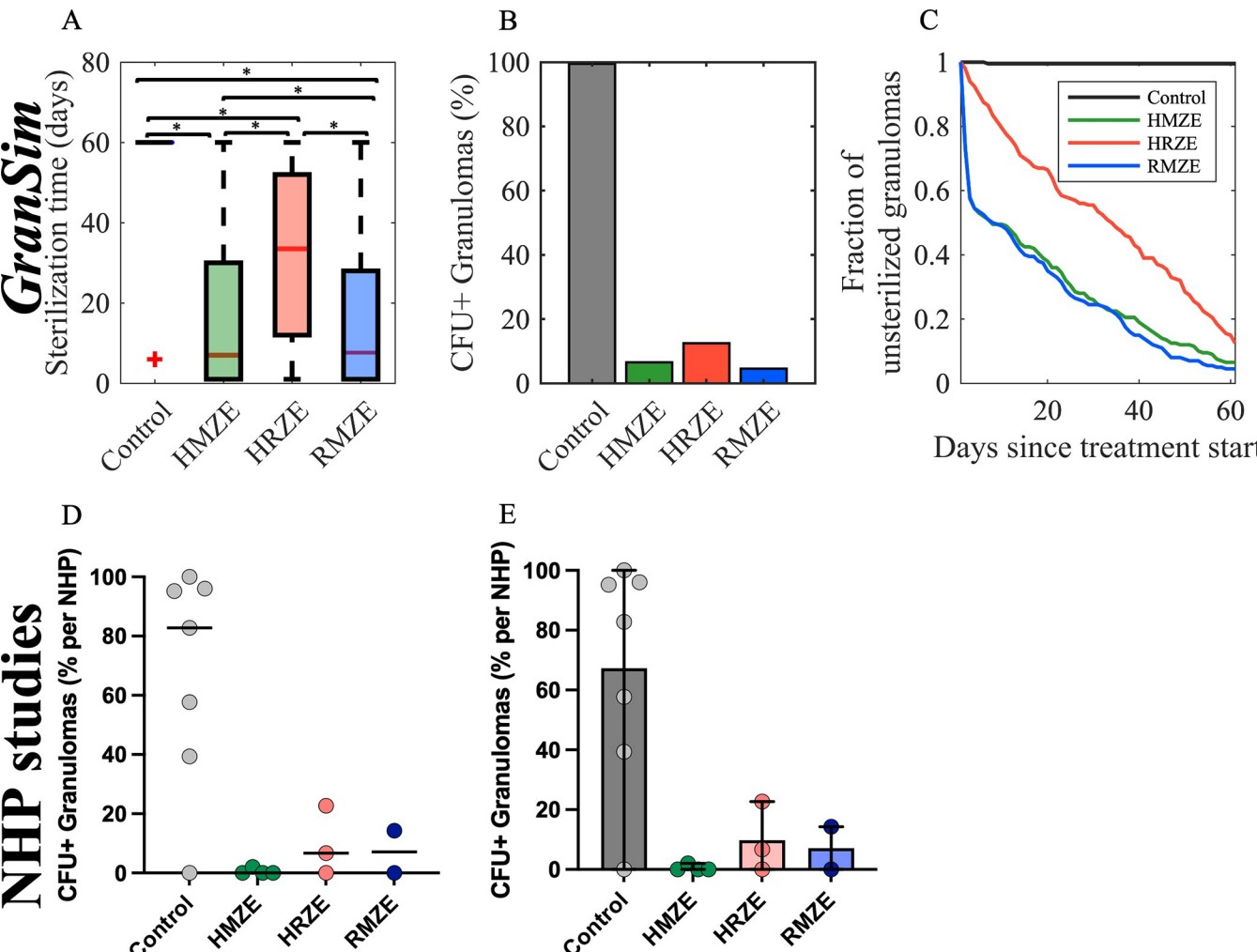

**Fig 3. Comparing NHP and *GranSim* regimens.** Comparison of the standard regimen (HRZE) to two moxifloxacin–containing regimens, HMZE and RMZE, between *in silico* studies with *GranSim* (panels A–C) and *in vivo* NHP studies (panels D and E). (A) The sterilization times of granulomas averaged over 200 granulomas in *GranSim*. Note that we assign the maximum simulation time of 60 days as a sterilization time for unsterilized granulomas (*p<0.001, one–tailed paired t–test). The central red lines in box plots represent the median, whereas the bottom and the top edges of boxes represent 25th and 75th percentiles, respectively. (B, E) Percentage of granulomas that are unsterilized by treatment end for (B) NHP studies and (E) *GranSim*. Colored dots in (E) represent the percentage of unsterilized granulomas per NHP. (C) The fraction of granulomas which are unsterilized as a function of simulated treatment time using *GranSim*. (D) The average total CFU per NHP after treatment with the corresponding regimens for two months (n = 7 animals in the control group, n = 3 animals in HRZE group, n = 4 animals in HMZE, n = 2 animals in RMZE). Statistical analyses were not performed on the NHP data due to small numbers of animals per group.

### Simulations reveal that moxifloxacin-containing regimens have a better bactericidal activity than HRZE

To systematically compare the efficacy of moxifloxacin-containing regimens to the standard regimen, we used *GranSim* to simulate treatment with all 4-way combinations of HRZEM (HRZE, RMZE, HMZE, HRME and HRZM) for 120 days (Fig 5). We analyze simulation results distinguishing granulomas that are high-CFU (Fig 5A and 5D) versus low-CFU (Fig 5B and 5E), as well as combined (Fig 5C and 5F). We used equal sample sizes (N = 100) for high- and low-CFU granulomas for a fairer comparison. However, data from NHP granulomas (black dots in Fig 1) suggest that 30% of unsterilized granulomas are high-CFU, while 70% of them are low-CFU granulomas. Although low-CFU granulomas are clinically more common

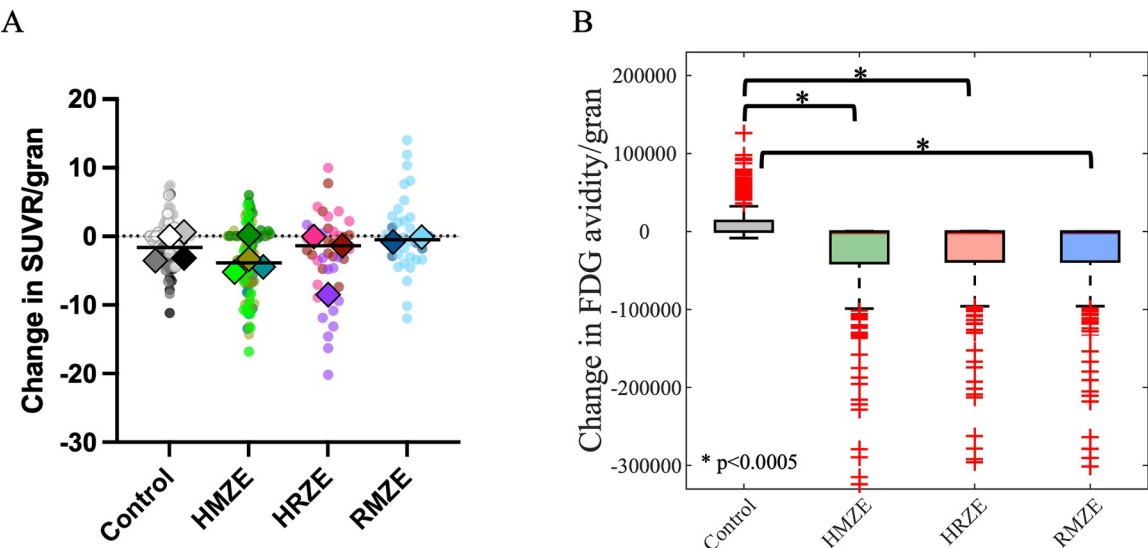

**Fig 4. Comparison of metabolic activity (measured by SUVR) change post treatment in NHP and *GranSim*.** Comparison of metabolic activity changes (A) in NHP granulomas and (B) using *GranSim*. (A) Change in standardized uptake value ratio (SUVR) per NHP granuloma (colored dots) in 8 weeks ($SUVR_{8weeks} - SUVR_{pre-treatment}$) when NHP are treated with HRZE (n = 3 animals), HMZE (n = 4 animals) and RMZE (n = 2 animals) for 8 weeks (n = 7 animals in control case, i.e., without treatment). Color shades of the dots in each column indicate NHPs and the diamonds are the median of SUVR change/granuloma for each NHP. (B) Change in FDG avidity per granuloma simulated using *GranSim* (FDG avidity$_{8weeks}$−FDG avidity$_{pretreatment}$) averaged over 200 granulomas (*p<0.0005, one–tailed paired t–test). The central red lines in box plots represent the median, whereas the bottom and the top edges of boxes represent $25^{th}$ and $75^{th}$ percentiles, respectively.

as most individuals have latent infection, high-CFU granulomas are harder to treat and likely result in an active disease. Therefore, it is crucial to understand drug dynamics within high-CFU granulomas.

Our simulation results indicate that all four regimens containing moxifloxacin clear Mtb within all types of granulomas (high-CFU, low-CFU, and combinations) in a significantly shorter time than the standard regimen HRZE (blue curve in Fig 5A–5C, gray box in Fig 5D–5F). Moreover, simulations show that the initial decline in bacterial load for combinations of high- and low-CFU granulomas with regimens containing moxifloxacin (Fig 5C) stems from the fast sterilization of all low-CFU granulomas (Fig 5B), as the clearance rate for high-CFU granulomas is slower than that for low-CFU granulomas. In addition, the differences between various moxifloxacin-containing regimens are more pronounced in high-CFU granulomas. For example, HRZM clears all high-CFU granulomas by 51 days, which is the fastest of all 4-way combinations of HRZEM. The next best regimen is HRME, requiring 77 days to sterilize all granulomas with high-CFU. RMZE and HRZE sterilize all high-CFU granulomas by a similar time window, in 94 and 97 days, respectively. Lastly, treating all granulomas until they sterilize with HMZE takes 118 days. However, the time required to sterilize a high-CFU granuloma (Fig 5D) is lower for moxifloxacin-containing regimens (Fig 5D, red boxes) than that for the standard regimen (Fig 5D, black box), which is consistent with the findings in Figs 2 and 3.

## Simulations show that moxifloxacin-containing regimens are more efficacious than HRZE with fewer than four antibiotics

Compliance is one of the challenges of TB treatment due to the long-term use of many antibiotics with numerous side effects. To identify a more patient-friendly treatment, in line with the

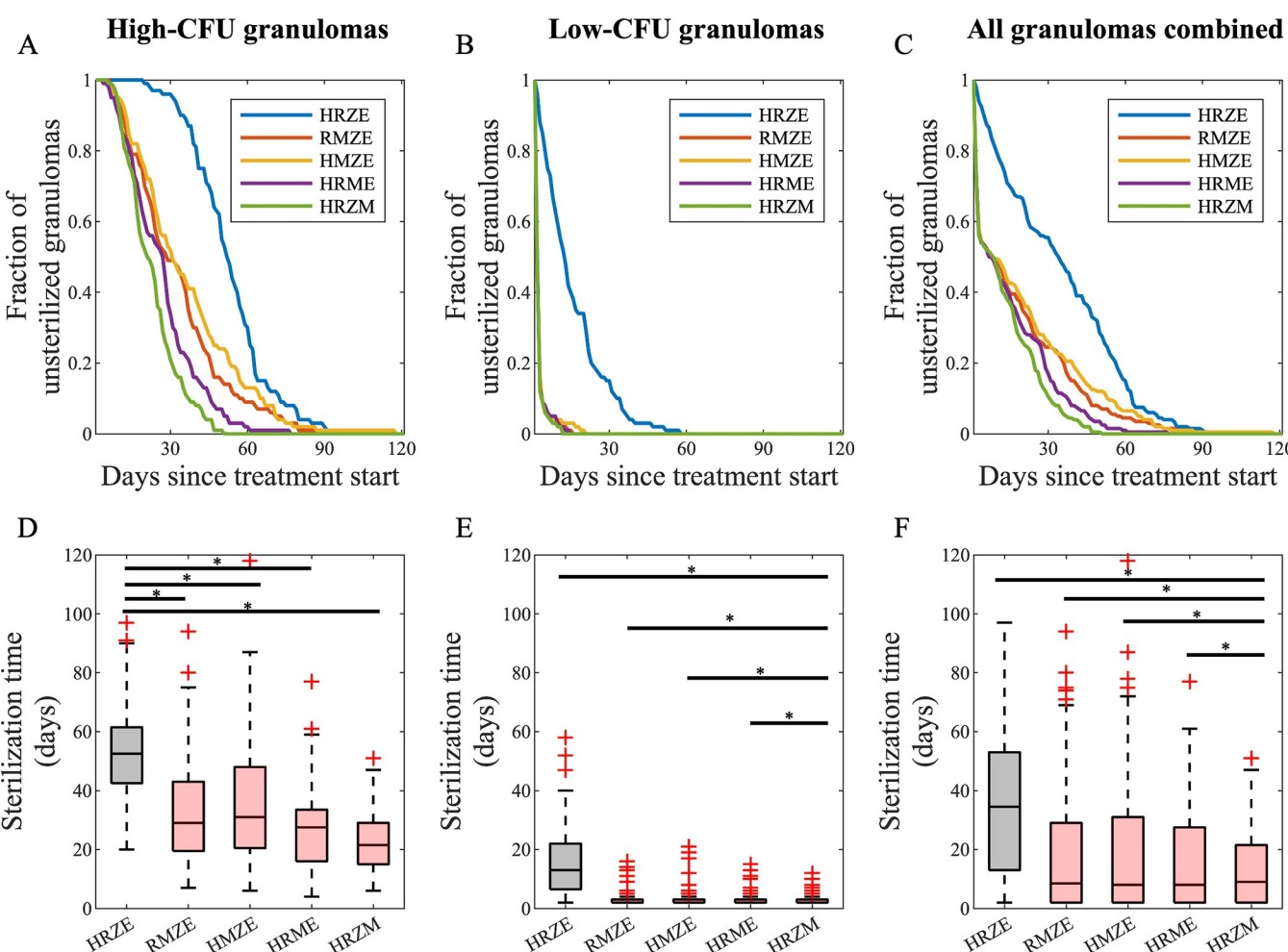

**Fig 5. Comparison of 200 simulations for all HRZEM four–way regimens using *GranSim*.** Comparison of (A–C) sterilizing rates and (D–F) sterilization times of 4–way combinations of HRZEM for (A and D) 100 high–CFU, (B and E) 100 low–CFU and (C and F) a combination of 100 high–and 100 low–CFU granulomas. Significance test was performed only between HRZE and moxifloxacin–containing regimens (*p<0.0001, one–tailed paired t–test). The central lines in box plots represent the median, whereas the bottom and the top edges of boxes represent 25th and 75th percentiles, respectively.

goals of the END TB strategy of WHO [53], we could reduce the number of antibiotics used in a regimen and/or reduce the total dose of a regimen. To test whether a regimen with fewer than four antibiotics would be as efficient as (or more efficient than) the 4-way combinations of HRZEM, we simulated all 3-way (Fig C in S1 Appendix) and 2-way (Fig D in S1 Appendix) combinations of HRZEM in treating individual granulomas. As compared with 4-way combinations (Fig 5B and 5E), we also observed the fast clearance of low-CFU granulomas treated with moxifloxacin-containing regimens in 3- (panels B and E in Fig C in S1 Appendix) and 2-way (panels B and E in Fig D in S1 Appendix) combinations. Sterilization of high-CFU granulomas remains faster with 3-way combinations containing moxifloxacin than for regimens without moxifloxacin; however, the rate of sterilization is slower than for low-CFU granulomas (panels A and D in Fig C in S1 Appendix). The trend does not always hold for treatment of high-CFU granulomas with 2-way combinations containing moxifloxacin (panels A and D in Fig D in S1 Appendix). Regimens like ZM and EM cannot sterilize most of the high-CFU granulomas despite prolonged treatment (panel A in Fig D in S1 Appendix). These

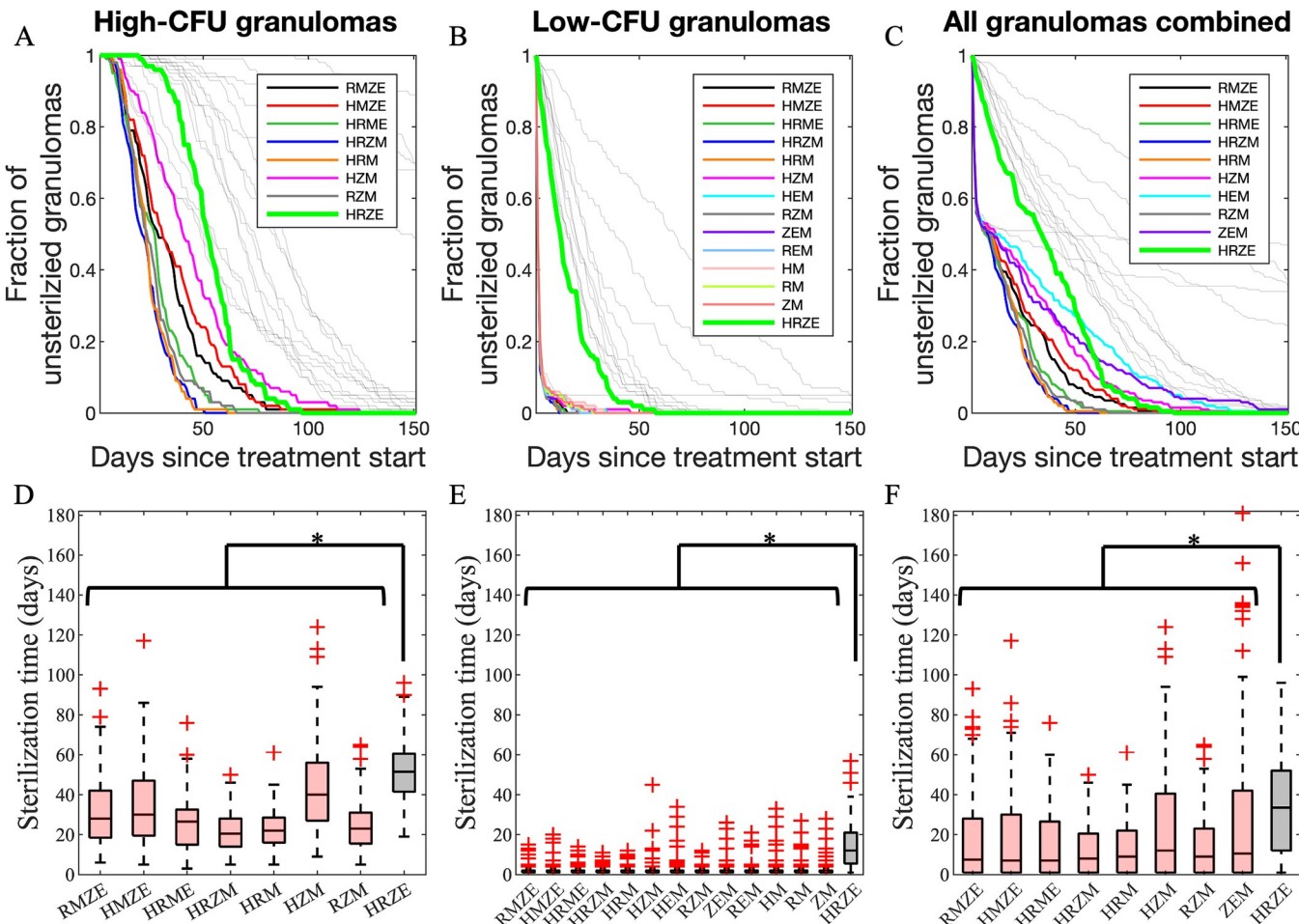

**Fig 6. Simulated treatments with 4–way, 3–way, 2–way regimen comparison from HRZEM.** Comparison of (A–C) sterilizing rates and (D–F) sterilization times of all regimen combinations of HRZEM to HRZE (thick green curve in all panels) in (A and D) 100 high–CFU, (B and E) 100 low–CFU and (C and F) a combination of 100 high–and 100 low–CFU granulomas. Significance test was performed only between HRZE and all other regimens (*p<0.01, one–tailed paired t–test). The central lines in box plots represent the median, whereas the bottom and the top edges of boxes represent 25[th] and 75[th] percentiles, respectively. Colored curves indicate the regimens that clear granulomas faster than HRZE, i.e., they have a significantly lower sterilization times. Gray curves represent regimens with slower sterilization.

granulomas may be related to the classically defined paucibacillary granulomas which even after treatment remain difficult to sterilize [54].

Lastly, we compared treatments with all 2-way, 3-way and 4-way combinations of HRZEM to the standard regimen HRZE based on the sterilization time for each regimen of high-CFU (Fig 6A) and low-CFU (Fig 6B) granulomas and both types of granulomas combined (Fig 6C). Our results demonstrate that regimens that are more effective in sterilizing granulomas than HRZE each contain moxifloxacin (colored curves in Fig 6). For high-CFU granulomas, a moxifloxacin-containing regimen with at least 3 antibiotics is needed to achieve a better performance than HRZE (Fig 6A). However, sterilizing low-CFU granulomas faster than HRZE is possible even with regimens containing two antibiotics (HM, RM and ZM in Fig 6B). These comparisons are based only on the standard doses of regimens; optimization of doses is also possible. Our results agree with preclinical studies on mouse models in that HRZM [40, 55], RMZE [55] and RMZ [39, 40] sterilize mice faster than HRZE.

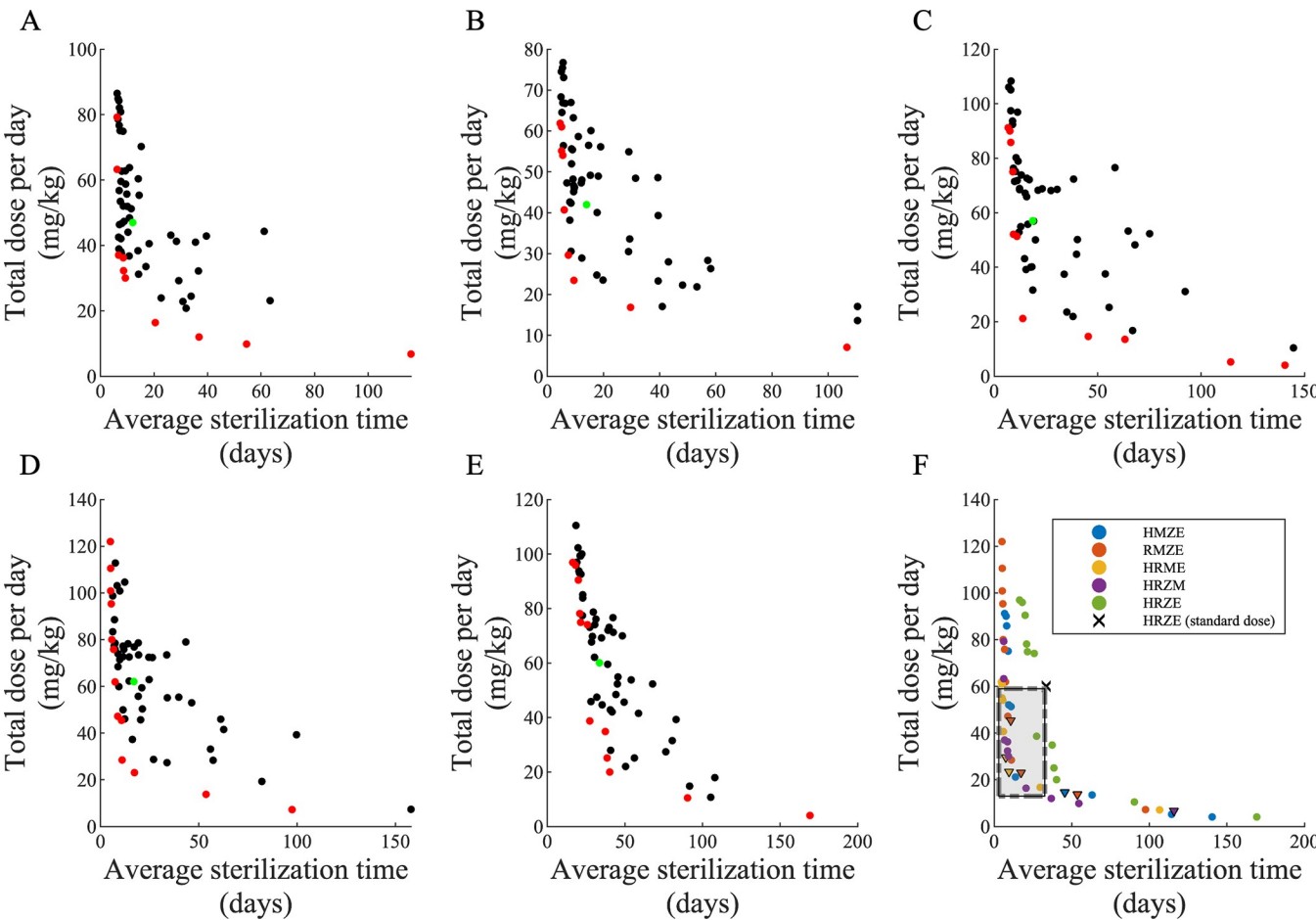

**Fig 7. Pareto front optimization study simulating all 4–way combinations of HRZEM to find regimens that minimize both average sterilization time and total dose.** Pareto front optimization identifying optimal dose and sterilization times for: (A) HRZM, (B) HRME, (C) HMZE, (D) RMZE and (E) HRZE. In each panel (A–E), red dots represent the (non–dominating) regimens that belong to the Pareto set (see Tables B–F in S1 Appendix for the doses of each antibiotic in the regimens that belong to Pareto sets) whereas black dots are the regimens that are not optimal based on the objectives. Green dots show the regimen based on the current standard doses recommended by CDC [4]. (F) Pareto sets for all regimens (same as red dots in panels A–E) compared to the standard regimen HRZE with CDC–recommended doses (**X** in Panel F). Dots in the dashed gray rectangle indicate the regimens that have lower total drug dose and lower average sterilization times (see Table 2 for the doses of each antibiotic in these regimens). Triangles indicate optimized regimens with 3–way combinations, as the optimal doses of one antibiotic (E or Z) in these regimens are predicted as 0.

## Dosing optimization method identifies new doses for regimens

Optimizing the dose of each antibiotic in all 4-way combinations of HRZEM may reduce the total antibiotic dose, contributing to our goal of a more patient-friendly TB treatment. Here, we use Pareto optimization to predict optimal solutions that balance the trade-off between two treatment objectives: minimizing the total antibiotic dose and minimizing the average time for regimens required to sterilize all Mtb within granulomas, i.e., the *average sterilization time*. Based on these two objectives, our Pareto optimization pipeline predicts the Pareto front for each 4-way combination regimen of HRZEM (HRZM, HRME, HMZE, RMZE and HRZE) and outputs a set of optimized regimens that belong to the Pareto set (i.e., optimal doses) (Fig 7A-E, red dots).

In general, regimens that simulations identify as optimal (i.e., regimens in the Pareto set) span a wide range of total dose and average sterilization times. This suggests that among optimized regimens, some have very low average sterilization times at the cost of a very high antibiotic dose and some regimens have a very low total dose leading to long sterilization times.

**Table 2. Simulated Doses of Antibiotics that optimize treatment objectives (compare with Fig 7).** The doses for each antibiotic in the regimens that have lower average sterilization time and lower dosage than the standard regimen (black row) as shown in Fig 7F (i.e., all regimens in dashed gray rectangle). The underlined rows indicate optimal 3–way combinations, where the optimal dose of E or Z is predicted as 0.

| Avg Sterilization Time (days) | Total dose (mg/kg) | H | R | Z | E | M |
|---|---|---|---|---|---|---|
| **33.6** | **60** | **5** | **10** | **25** | **20** | **0** |
| 5.1 | 55.2 | 6.1 | 20.0 | 0 | 16.8 | 12.3 |
| 5.5 | 53.4 | 7.5 | 13.9 | 0 | 19.7 | 12.3 |
| 6.1 | 40.8 | 9.0 | 11.3 | 0 | 7.2 | 13.3 |
| 6.7 | 37.0 | 8.8 | 19.7 | 0.6 | 0 | 7.9 |
| _7.5_ | _29.6_ | _5.5_ | _10.1_ | _0_ | _0_ | _14.0_ |
| 8.6 | 36.4 | 1.3 | 16.0 | 9.0 | 0 | 10.1 |
| 8.7 | 32.3 | 8.7 | 7.7 | 5.1 | 0 | 10.8 |
| 8.8 | 47.2 | 0 | 18.4 | 16.0 | 4.1 | 8.7 |
| 9.2 | 30.1 | 10 | 9.7 | 3.9 | 0 | 6.5 |
| 9.3 | 52.2 | 8.7 | 0 | 31.2 | 0.5 | 11.8 |
| _9.5_ | _23.4_ | _6.7_ | _9.2_ | _0_ | _0_ | _7.5_ |
| _10.6_ | _45.4_ | _0_ | _9.2_ | _26.9_ | _0_ | _9.3_ |
| 10.9 | 51.3 | 9.8 | 0 | 18.7 | 12.9 | 9.9 |
| 11.0 | 28.4 | 0 | 13.3 | 2.6 | 1.0 | 11.5 |
| 13.8 | 21.1 | 8.2 | 0 | 2.6 | 1.0 | 9.3 |
| _17.3_ | _23.0_ | _0_ | _12.7_ | _0_ | _2.5_ | _7.8_ |
| 20.5 | 16.4 | 9.1 | 0.9 | 1.2 | 0 | 5.2 |
| 27.3 | 38.7 | 4.7 | 19.0 | 14.2 | 0.8 | 0 |
| 29.7 | 16.8 | 5.4 | 5.1 | 0 | 3.1 | 3.2 |

However, we are particularly interested in optimized regimens that have both lower total dose and lower average sterilization times as compared to standard regimen (HRZE with CDC-recommended doses). Our method predicts that the 19 regimens in the dashed gray rectangle (Fig 7F) are all more advantageous than the standard regimen (black dot in Fig 7F) in terms of reducing *both* total dose and sterilization time. These regimens tend to have higher doses of rifampicin than the standard regimen yet lower total regimen dose (Table 2), resulting in shorter sterilization times, which is in line with clinical trials that showed a reduction in time to culture conversion using higher doses of rifampicin [56,57]. Based on our earlier results, it is not surprising that these optimized regimens mostly contain moxifloxacin (Table 2). This is also expected based on clinical studies where moxifloxacin-containing regimens sterilize granulomas more efficiently (c.f. Fig 2). Further, although most of these optimal regimens contain four antibiotics, our pipeline also predicted a few optimal combinations with less than four antibiotics (see triangles in the rectangle region of Fig 7F; underlined rows labeled in Table 2). (Our pipeline predicts the ethambutol optimal dose as 0 for HRME and RMZE regimens and the pyrazinamide optimal dose as 0 for RMZE, thus identifying HRM, RMZ and REM as additional optimal regimens). This agrees with our systematic study of all possible combinations that determined HRM, RZM and REM as more efficient regimens than the standard regimen HRZE (c.f. Fig 6). Our optimization approach provides a more efficient way to identify regimens with different combinations of antibiotics than is possible in clinical or experimental studies.

## Discussion

One of the strategies to improve TB treatment regimens is to shorten treatment duration by introducing or substituting newer antibiotics that have better bactericidal activities than those

in the current standard regimen (HRZE), for example, by considering bedaquiline, pretomanid, linezolid [58] or moxifloxacin [33]. To this end, clinical trials [33,41,43] and also *in vivo* studies with mice [39,40,44] have been conducted to explore a moxifloxacin-containing regimen that decreases treatment duration. In this study, we employ three unique approaches to predict more patient-friendly treatment regimens for TB: replacing antibiotics in the standard regimen with moxifloxacin *in vivo* and *in silico*, reducing the total number of antibiotics in a regimen by scanning all regimen combinations *in silico*, and reducing the total dosage using an *in silico* drug optimization pipeline.

Previously, we explored regimens with and without moxifloxacin in our simulation framework, *GranSim*, to compare with early-phase clinical trials [59]. However, this is the first study that directly compares TB treatment simulations using *GranSim* to a phase 3 clinical trial (REMoxTB [33]) (Fig 2). We also perform NHP studies with promising regimens predicted by *GranSim*. (Figs 3 and 4). Different from our previous studies, we systematically analyzed all possible combinations with or without moxifloxacin and employed a new optimization pipeline to identify optimal regimens that sterilize granulomas more efficiently than HRZE.

Previous clinical trials concluded that both 4 months of HRZM treatment and 4 months of RMZE had better bactericidal activity than the control group (HRZE) based on the conversion to culture negativity status of the patients (Fig 2A) [33]. In our simulations, we observe a similar trend as in the clinical trial in terms of bactericidal activity: HRZM and RMZE groups are more effective in reducing bacterial load and sterilizing granulomas than the control group (Fig 2B and 2C). Although the measures of the clinical trial and our simulations are at different scales (host-scale measures in clinical trials and granuloma-scale measures in simulations), both studies support that moxifloxacin is a promising regimen.

NHP experiments with standard and moxifloxacin-containing regimens indicate that all regimens reduce the total CFU of NHPs (Fig 3D) by sterilizing the majority of NHP granulomas (Fig 3E and Fig E in S1 Appendix). Furthermore, granuloma metabolic activity (surrogate for inflammation) drops by treatment with HMZE only, while RMZE and HRZE does not affect the metabolic activity. These outcomes are in agreement with our simulations; however, simulations predict that RMZE is equally effective as HMZE in sterilizing granulomas and reduces metabolic activity. This difference may stem from the small sample size in NHP studies but likely also is due to *in vivo* factors that are not included within *GranSim*. In addition, the surrogate measures used for FDG avidity in granulomas are only an approximation of the factors involved in *in vivo* FDG avidity. Using *GranSim*, we simulated 200 distinct granulomas per regimen. However, due to resource limitations, sample sizes were necessarily smaller in the NHP studies, and RMZE has the smallest sample size with only 2 animals. Moreover, unlike *in silico* studies where we simulate treatment with the same set of granulomas over various regimens, *in vivo* studies require different sets of animals to test regimens, and the outbred nature of macaques adds another level of variability, although this is also true in humans.

To test the efficacy of moxifloxacin-containing regimens more systematically and to potentially reduce the number of antibiotics needed per regimen, we simulated the treatment of low- and high-CFU granulomas with all 4-way (Fig 5), 3–way (Fig C in S1 Appendix) and 2-way (Fig D in S1 Appendix) combinations of HRZEM. In this study, we conclude that any 4-way, 3-way or 2-way (except EM) combinations that include moxifloxacin are more efficacious in eliminating bacteria within low-CFU granulomas than HRZE (Fig 6B). However, only 4-way combinations and some of the 3-way combinations work better than HRZE for treating high-CFU granulomas (Fig 6A). This suggests that decreasing the number of antibiotics within a regimen may be challenging when treating more progressive, caseous granulomas with the 5 drugs in this study, whereas granulomas with lower CFU numbers are easier to treat with fewer antibiotics.

Most of the regimens that are 2-, 3- or 4-way combinations of HRZEM consistently decrease the fraction of granulomas remaining unsterilized over the course of the treatment and, subsequently, clear them all (Fig 5A–5C, panels A-C in Fig C and Fig D in S1 Appendix). However, some regimens cannot sterilize further, i.e., fraction of unsterilized granulomas stays the same over a prolonged treatment time, (especially high-CFU granulomas). This may follow as high-CFU granulomas are mainly caseous with bacteria trapped within that region. It is known that moxifloxacin does not homogeneously diffuse into the caseous core of granulomas [5,60]. Therefore, a regimen containing moxifloxacin needs to be complemented with antibiotics that are effective in killing Mtb trapped within caseum of granulomas, unlike ethambutol (E) [61,62] or pyrazinamide (Z) [20,61]. Treatment of high-CFU granulomas with ZM or EM decreases the bacterial load initially, but eventually results in granulomas with primarily Mtb in caseum (Fig F in S1 Appendix) that could not be sterilized by prolonged treatments with ZM (panel A in Fig F in S1 Appendix) or EM (panel B in Fig F in S1 Appendix). This suggests that ZM or EM treatments may result in granulomas that harbor bacteria that could later lead to relapse disease [63]. Although treatment with EM or ZM could not sterilize all high-CFU granulomas, 20% and 40% of high-CFU granulomas are cleared by EM and ZM treatment, respectively (panel A in Fig D in S1 Appendix).

Another novel approach introduced in this study is implementing an optimization pipeline into *GranSim* to optimize doses of the drugs within regimens using a multi-objective optimization algorithm. Previously, we studied optimization in *GranSim* by comparing genetic algorithm and radial basis function (RBF) network surrogate models and showed that using an RBF network method is more efficient in optimizing drug regimens without losing accuracy [37]. However, the RBF network method is based on minimizing one objective function that may consist of various weighted terms, depending on the objectives we consider to discover a better regimen. A Pareto set is a set of solutions that is used to minimize multiple objectives with varying levels of importance. Therefore, determining the Pareto front with a single objective function would require many iterations of optimization to obtain a wide-ranging Pareto set [64]. Thus, multi-objective Pareto optimization is a more efficient approach to discover the optimal solution set.

To successfully optimize the doses of a regimen, it is crucial to have well defined objectives based on the factors that we would like to consider in a regimen. In this study, we assumed that the total dosage of a regimen, independent of the antibiotic, should be minimal. Moreover, the regimen should sterilize granulomas as quickly as possible. However, we can modify these objectives or add additional ones to obtain more biologically relevant optimized regimens. For instance, each antibiotic has different levels of adverse side effects and high doses should be avoided. Moreover, financial burden of each regimen should also be considered in order to identify more accessible treatments worldwide.

Computational modeling studies have many advantages that are useful for drug discovery studies and that complement experimental studies. Unlike clinical trials and *in vivo* experiments, our computational approach has the power to evaluate the efficacy of regimens on *the same set of granulomas to eliminate the variability*. Moreover, due to limited resources, trying every single regimen combination *in vivo* experimentally or clinically, or repeating the experiment many times to achieve significance may not be feasible. Hence, promising regimens may be skipped or missed due to nonsignificant results. Here, we predicted new and promising combination regimens that have not yet been studied clinically, such as HMZE that informed our NHP studies and was predicted to be an effective regimen via our simulations. Another drawback of clinical and *in vivo* studies is the risk of disease relapse. To assess the relapse rate after treatment, study subjects are observed for several months. Unlike experiments, we can track each Mtb bacilli in our simulations that gives us the power to anticipate relapse at the

end of the treatment based on the sterilization status. Having unsterilized granulomas at the end of treatment is predictive of TB relapse.

One limitation of our approach is that our model is at granuloma-scale. However, predicting the relapse rates requires a host-scale model. Additionally being able to treat a collection of granulomas within a host can serve to elaborate further the studies herein. The source of relapse is not fully understood. One hypothesis suggests that bacteria within granulomas in lymph nodes could migrate to the lungs and induce reinfection or reactivation [51]. Therefore, a host-scale immune model of TB that contains multiple granulomas within lungs and lymph nodes is needed to assess regimens' long-term efficacy and to determine relapse rates, which are crucial parameters to evaluate regimens efficiently, and we are currently adapting our host-scale TB model, *HostSim* [46, 65], to encompass antibiotics and meet this need. These next-generation improvements will make our approach more powerful and reliable, so that *in vivo* experiments or clinical trials can be systematically informed by simulation results. In current and future work, we aim to test the most efficacious regimens in an animal model to validate our predictions.

## Methods

We combined computational modeling with studies in NHPs, outlining the approaches for each below. We point out where modifications to existing protocols and models have changed in the next-generation versions used herein.

### GranSim

As a basis for studying treatment at the granuloma scale, we used our well-established computational model of granuloma formation and function, *GranSim* (Fig 8B). *GranSim* is a hybrid agent-based model (ABM) that simulates the immune response during Mtb infection, capturing granuloma formation as an emergent behavior [6, 66–69]. Agents in this ABM include immune cells, such as macrophages and T-cells, and individual bacteria. *GranSim* simulates a two-dimensional section of lung tissue (6mm x 6mm) represented by dissecting a 300 x 300 grid into 90,000 grid microcompartments, each of size 20μm. Simulations begin with a single infected macrophage in the center of the grid that initiates recruitment of additional macrophages and T cells to the infection site. These immune cells interact with each other and with Mtb according to a large set of immunology-based rules that describe killing of Mtb, secretion of chemokines/cytokines, and activation and movement of cells (for a complete description of our rules, see http://malthus.micro.med.umich.edu/GranSim/). Granulomas "emerge" as a result of these interactions when simulating *GranSim*. Infection is initiated with a single bacterium.

NHPs are highly informative animal models for TB, as TB disease and pathology, including granulomas, are similar to humans [70]. The immunological rules and cellular behaviors included in *GranSim* are based on datasets derived from NHP granulomas [66,67,69]. Moreover, we validate and calibrate *GranSim* granulomas to both spatial and temporal datasets from NHP granulomas, including immune cells (macrophages and T cells) and Mtb counts and the spatial distribution of cell types within a granuloma [30,50,71]. *GranSim* simulates a broad range of biologically relevant outcomes that can recreate the heterogeneity of observed granulomas from NHPs and humans [67,72].

Antibiotics may have bactericidal (bacterial killing) or bacteriostatic (inhibition of bacterial growth) effects. To capture the actions of these drugs on bacteria, tracking the individual bacteria within the granuloma is key [73]. To mimic that of actual infection in NHPs, we simulate three distinct subpopulations of Mtb based on their location within granulomas: replicating-

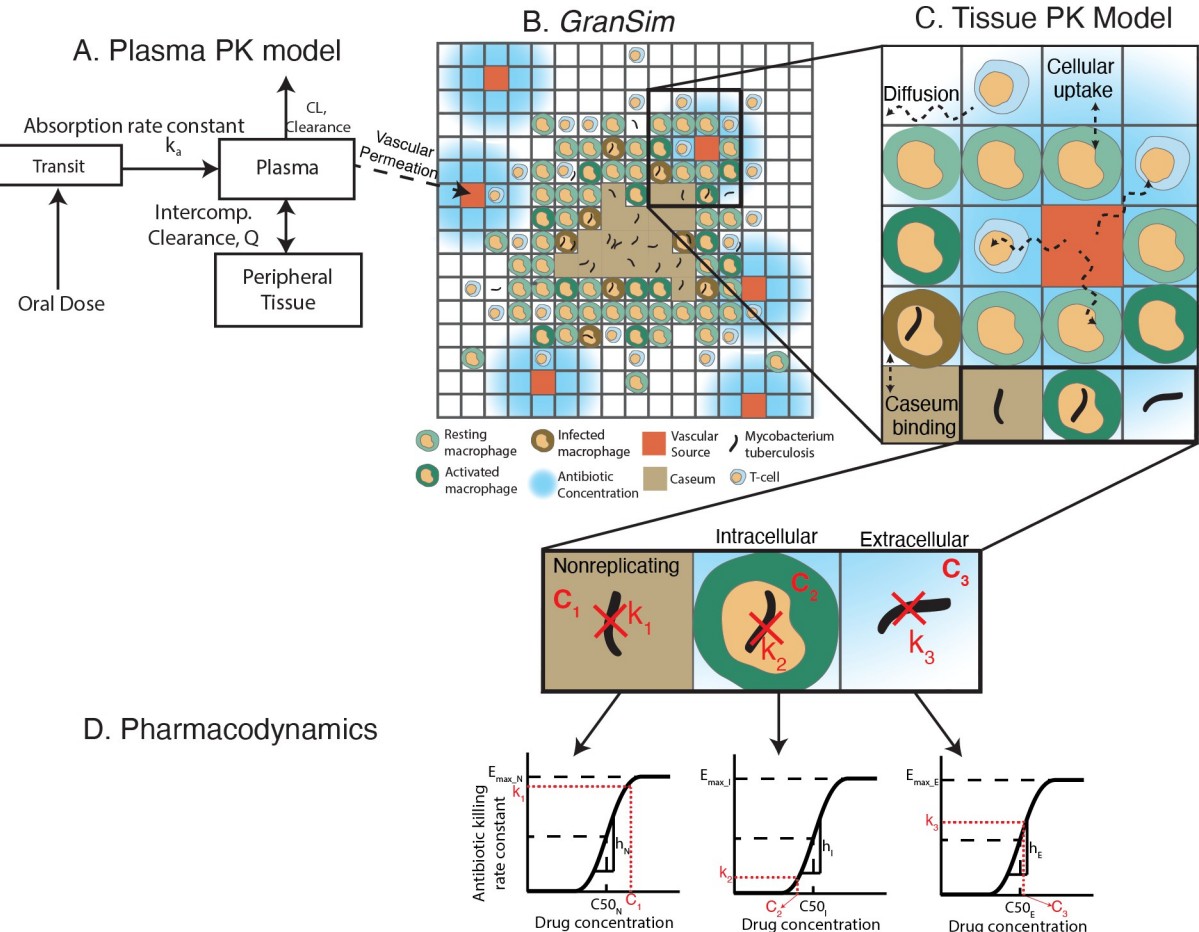

**Fig 8. An overview of the hybrid agent–based model that simulates granuloma formation and function, *GranSim*, and how pharmacokinetics (PK) and pharmacodynamics (PD) of antibiotics are incorporated in *GranSim*.** A) Our simulations begin with *GranSim* generating a large library of granulomas to be used for regimen testing. B) Antibiotic concentrations in plasma are simulated by a compartmental, ordinary differential equation model ($k_a$, CL and Q are rate constants between compartments) with one (Z, E and M) or two transit compartments (H and R) representing oral absorption (one transit compartment shown in the figure). (C) Antibiotics in the plasma permeate through vascular sources into the lung tissue, i.e., onto the spatial grid of *GranSim*, where antibiotics can: diffuse, bind to caseum, be taken up by macrophages and penetrate into a granuloma. (D) We determine a killing rate for each Mtb phenotype ($k_1$, $k_2$ and $k_3$) based on the local antibiotic concentration in their environment (grid compartment) ($C_1$, $C_2$ and $C_3$) using a Hill equation calibrated to each Mtb type (nonreplicating, intracellular and extracellular). $E_{max\_N}$, $E_{max\_I}$ and $E_{max\_E}$ are maximum–killing rate constants, respectively, $C50_N$, $C50_I$ and $C50_E$ are the concentration at half maximal killing, respectively and $h_N$, $h_I$ and $h_E$ are Hill curve constants for nonreplicating, intracellular and extracellular Mtb, respectively.

extracellular Mtb, intracellular Mtb that reside and replicate within macrophages, and Mtb that are trapped within the caseous necrotic core. These caseum-trapped bacteria have varying growth rates depending on the level of tissue caseation. These subpopulations differ in their abilities to replicate and move within a granuloma.

## Latin Hypercube Sampling (LHS)

LHS is a parameter-sampling method that samples the parameter space without replacement and covers the parameter space more uniformly than a simple random sampling. It is done by dividing each parameter distribution into N equal probability intervals and sampling from these intervals to generate *N* distinct parameter sets and identify epistemic uncertainty [48,74,75]. We used this method to generate an *in silico* library of granulomas in *GranSim*. If

the system under study, as is ours, has stochastic components, it is necessary to do replicates (we choose 3) of each of the $N$ runs to capture the aleatory uncertainty within as well (c.f. [48]). These samplings capture both epistemic and aleatory uncertainty that arise in parameter sets.

## Pharmacokinetic/pharmacodynamic (PK/PD) modeling

We have used *GranSim* previously to simulate the PK/PD of antibiotic drug treatment for TB. Specifically, we can simulate the spatial distribution of antibiotics and their sterilizing ability for different antibiotic regimens [5,6,8,38].

Briefly, the PK/PD model within *GranSim* simulates the plasma concentration over time following oral doses of antibiotics (Fig 8A), the subsequent spatial concentration in the simulated granuloma (Fig 8C) and the bactericidal activity based on the local concentration (Fig 8D). We modeled the plasma PK using a compartmental, ordinary differential equation model to simulate absorption through transit compartments into the plasma, exchange with peripheral tissue and first-order elimination from the plasma (Fig 8A) [6,22]. To simulate tissue PK, we referenced the concentration in the plasma and calculated flux through vascular sources on the computational grid. We then calculated diffusion through tissue, binding to caseum and epithelium and partitioning into macrophages (Fig 8C) [6,8,38,76]. These processes control penetration depth, i.e., how deeply antibiotics penetrate into granulomas.

We modeled the PD by using a Hill function that determines the concentration ($C$) dependent antibiotic killing rate constant ($k$), which is the rate of bacterial death per time step [77] (Fig 8D). The general form of the Hill curve we use is:

$$k(C) = E_{max} \frac{C^h}{C^h + C_{50}{}^h} \tag{Eq1}$$

where $E_{max}$ is the maximum killing rate constant, $h$ is the Hill coefficient and $C_{50}$ is the concentration needed to achieve the half maximal killing rate constant ($E_{max}/2$). For each antibiotic, we calibrated the parameters of the Hill curve ($E_{max}$, $C_{50}$ and $h$) for intracellular, replicating-extracellular and caseum Mtb separately, as the pharmacodynamics of antibiotics are different in these subpopulations. The calibration is based on bactericidal assays of Mtb within microenvironments that are relevant to NHP granulomas, namely infected macrophages [18,19,78–80], Mtb in rich growth media [18,19,78–80] and Mtb in caseum mimic [20], for intracellular, replicating-extracellular and Mtb in caseum, respectively (Fig 8D).

In this study, we used the effective concentration of each antibiotic ($C$) as the total concentration in each grid compartment rather than the free concentration, i.e., the extracellular concentration that is not bound to any macromolecules or any tissue, as we calculated in our previous studies [8, 59]. We made this change as the bactericidal assays we reference are based on the total concentration applied to the Mtb *in vitro* [20].

## Accounting for pharmacodynamic drug interactions in the model

When multiple antibiotics are used and thus present and available on our simulation grid within *GranSim*, we simulate their interaction by adjusting the effective concentration according to their predicted fractional inhibitory concentration (FIC) values, as we have done previously [59]. We use the FIC values predicted by an *in silico* tool, INDIGO-MTB (inferring drug interactions using chemogenomics and orthology optimized for Mtb) [31,32]. This tool employs a machine learning algorithm that uses known drug interactions along with drug transcriptomics data as inputs and predicts unknown drug interactions, i.e., FICs.

Briefly, we first converted the concentrations of all antibiotics on a small section of the grid (microgrid) to the equipotent concentration of the antibiotic of the highest maximal killing

rate constant (highest $E_{max}$). For example, if we have n antibiotics (drug $i$ with the concentration $C_i$) and drug m has the highest $E_{max}$ of all drugs, then we calculate the adjusted concentration for each drug $i$ ($C_{i,adj}$), which is the concentration of drug m that results in the same antibiotic killing rate constant as drug $i$ with the concentration of $C_i$, with the following equation:

$$C_{i,adj} = \left( \frac{C_{m,50}{}^{h_m} C_i{}^{h_i}}{\frac{E_{max,m}}{E_{max,i}} \left( C_i{}^{h_i} + C_{i,50}{}^{h_i} \right) - C_i{}^{h_i}} \right)^{1/h_m} \qquad \text{(Eq2)}$$

where $C_{m,50}$ and $C_{i,50}$ are the concentration of $C_m$ and $C_i$ at which half maximal killing is achieved, respectively, $E_{max,m}$ and $E_{max,i}$ are the maximal killing rate constants of drug m and drug $i$, respectively, and $h_m$ and $h_i$ are the Hill coefficients of drug m and drug $i$, respectively. Then, we calculated the effective concentration ($C_{eff}$) as the sum of the adjusted concentrations of $n$ antibiotics that are increased/decreased based on the FIC values (see Table A in S1 Appendix for a complete list of FIC values) to simulate synergistic/antagonistic effects with the following equation:

$$C_{eff} = \left( \sum_{i=1}^{n} C_{i,adj}{}^{FIC} \right)^{1/FIC} \qquad \text{(Eq3)}$$

where $C_{i,adj}$ is the adjusted concentration of the drug $i$. Then, we used $C_{eff}$ to calculate the antibiotic killing rate constant $k$ on that microgrid by using the Hill equation constants of the antibiotic with the highest $E_{max}$:

$$k \left( C_{eff} \right) = E_{max,m} \frac{C_{eff}{}^{h_m}}{C_{eff}{}^{h_m} + C_{m,50}{}^{h_m}} \qquad \text{(Eq4)}$$

where $E_{max,m}$, $h_m$ and $C_{m,50}$ are the Hill equation parameters of the antibiotic m, the one with the highest $E_{max}$ within the regimen.

## Simulating antibiotic regimens in *GranSim*

To calibrate the PK parameters of each antibiotic in *GranSim*, we used antibiotic concentrations at various time points and tissue types (e.g., plasma, uninvolved lung, caseum and lesion) from human or rabbit/NHP samples after administering human-equivalent doses (see Fig C in S1 Text for moxifloxacin PK calibration). We calibrated plasma and tissue PK parameters for isoniazid, rifampicin, pyrazinamide and moxifloxacin based on human data [21] and the parameters for ethambutol from rabbit samples [62] as human data are not available for this antibiotic. We also utilized MALDI-MS images from human [21] and rabbit [60,62] samples that show the spatial distribution of antibiotics within granulomas as a validation for our tissue PK calibration.

We simulated regimens on 200 randomly selected granulomas from our *in silico* granuloma library (100 low CFU and 100 high CFU granulomas). We employed different dosing protocols based on the studies shown in Table 1. First, we simulated the protocols for the REMoxTB clinical trial [33] using *GranSim*. There were 3 different groups in this study: control group (HRZE), HRZM group and RMZE group. To simulate the control group, we dosed granulomas with HRZE daily for 8 weeks, followed by 18 weeks of daily dosing of HR. We simulated HRZM and RMZE groups by dosing granulomas for 17 weeks daily with HRZM and RMZE, respectively, followed by 9 weeks of a placebo phase, i.e., 9 weeks of no antibiotics (Table 1).

To compare our results to NHP studies performed herein, we simulated the regimens HRZE, HMZE and RMZE for 60 days by dosing daily. We also simulated a positive control case with the same granulomas but with no antibiotics (Table 1). Additionally, we simulated

all 2-way, 3-way and 4-way combinations of HRZEM until all granulomas sterilize or reach a stable state, i.e., until the fraction of granulomas that are not sterilized doesn't change significantly over time (120 days for 4-way combinations (5 regimens), 220 days for 3-way combinations (10 regimens), 300 days for 2-way combinations (10 regimens)) (Table 1).

## Average sterilization time measurement

A regimen's efficacy depends on how fast it can clear all Mtb within a granuloma. Therefore, we measured the average time a regimen needs to clear a granuloma, i.e., *average sterilization time*, as a way to assess regimens' potency. The average sterilization time of a regimen $i$ ($t_{ster_i}$) is

$$t_{ster_i} = \frac{\sum_{k=1}^{n} t_{ster_{i_k}}}{n} \tag{Eq5}$$

where $n$ is the number of granulomas treated by $i$ and $t_{ster_{i_k}}$ is the time that granuloma $k$ is treated with $i$ until total Mtb within $k$ is zero. If a granuloma $k$ is not sterilized by a regimen $i$ at the end of the treatment, then we assign $t_{ster_{i_k}} = t_{treatment}$ where $t_{treatment}$ is the duration of the treatment.

## NHP granuloma FDG avidity measurement in *GranSim*

Positron Emission Tomography and Computed Tomography (PET-CT) scans are used to measure metabolic activity of granulomas within NHP by quantifying the uptake of a glucose analog FDG (2-deoxy-2-[18F]-fluoro-D-glucose) via a measure called SUVR (standardized uptake value ratio) [52]. As a proxy for capturing the SUVR per granuloma from NHP experiments within our computational model, we developed a surrogate measurement in *GranSim* that combines the amount of proinflammatory activity derived from both tumor necrosis factor (TNF) with activity of proinflammatory cells (such as activated T cells and macrophages) that we define as FDG avidity. This is a way to represent the metabolic activity in the *in silico* granulomas. Specifically, we calculate FDG avidity measure for each granuloma, $i$ as:

$$FDG\ avidity_i = \sum_{k=1}^{n} \left( TNF_k + M_{r_k} + 4M_{i_k} + 9M_{ci_k} + 6M_{a_k} + 3T_{gam_k} + 3T_{cyt_k} \right) \tag{Eq6}$$

where $n$ is the number of grid microcompartments of the agent-based model grid in a simulation, $TNF_k$ is the TNF concentration within the microgrid $k$ in pg/ml (with an upper bound as 30 pg/ml), $M_{r_k}$, $M_{i_k}$, $M_{ci_k}$, $M_{a_k}$, $T_{gam_k}$ and $T_{cyt_k}$ are the number of resting macrophages, infected macrophages, chronically infected macrophages, active macrophages, IFN-γ producing T cells and cytotoxic T cells at microgrid k, respectively (see Fig A in S1 Appendix for the visualization of FDG avidity in *GranSim* and see http://malthus.micro.med.umich.edu/GranSim/ for more information about the roles of each cell type). The weights that each cell type contributes to the FDG avidity on a grid is determined based on the assumed inflammatory responses each cell type creates based on their *in vivo* activity. Because we do not know all factors or cells that contribute to the SUVR, we use levels of TNF (an inflammatory marker) as a surrogate to represent contributions from other cells to the metabolic activity within a granuloma.

## Objective functions for regimen optimization

We use two objective functions to be minimized, the average sterilization time (described above) and the total normalized dose ($d$). We define the total normalized dose $d$ as

$$d(x) = \sum_{k=1}^{n} \frac{D_i}{D_{i_{max}}}, \tag{Eq7}$$

where $k$ is the number of antibiotics in the regimen $x$, $D_i$ stands for the dose of the individual antibiotic $i$, and $D_{i_{max}}$ is the maximum allowed dose in our simulations. Minimizing drug dose will decrease potential side effects. In our optimization pipeline, we aim to find the regimens that minimize both objective functions.

The sampling ranges for each dose variable were set to range from 0 mg/kg to double the standard CDC dose [4]. Maximum safe doses for each antibiotic were set to 10, 20, 40, 50 and 14 mg/kg for H, R, E, Z and M, respectively, as higher doses would increase the risk of toxicity and would not be clinically relevant [56,81–84].

## Kriging-based surrogate model

The goal of a multi-objective optimization is to find the optimal trade-off between two or more objectives by identifying the optimal variable combinations [85]. For example, in this study the goal is to both minimize time to sterilization and drug doses between multiple drugs. Using a surrogate-assisted framework involves predicting the objective functions based on the outcomes of the already-sampled regimens. These predictions can then be used as a computationally inexpensive alternative to predict the objective functions throughout the whole design space.

Here, we use a kriging-based surrogate model to generate the objective function predictions. This kriging-based prediction and optimization algorithm is based on a set of open-source MATLAB functions developed by Forrester and Sóbester [64,86]. This surrogate-assisted framework provides an efficient and accurate way to thoroughly investigate the regimen design space and predict optimal doses but with few iterations as compared to, for example, a genetic algorithm [37]. Based on the sampled regimens and the calculated values of the corresponding objective functions, the algorithm builds a kriging-based, surrogate model to predict the values of the objective functions at any point in the variable design space.

The kriging model operates by assuming that the value of a function $f$ of $n$ variables at any n-dimensional vector $x$ can be stated as the sum of some unknown mean ($\mu$) and an error term that is a function of position $\varepsilon(x)$ [87]:

$$f(x) = \mu + \varepsilon(x) \tag{Eq8}$$

We also assume that the error term $\varepsilon(x)$ is normally distributed with a mean of 0 and a standard deviation of $\sigma^2$. To provide an estimate for the error at any given $x$, we assume the errors at two points are correlated based on the distance between those two points. This means points that are closer in the variable space tend to be more related and have smaller variance than those that are farther. Hence, the correlation in error between points $i$ and $j$, equal to component $R_{ij}$ in the correlation matrix $R$, exponentially decays with respect to the weighted distance between them:

$$R_{ij} = Corr[\varepsilon(x^{(i)}), \varepsilon(x^{(j)})] = exp[-\sum_{h=1}^{n} \theta_h |x_h^{(i)} - x_h^{(j)}|^{p_h}](\theta_h \geq 0, p_h \in [1, 2]) \tag{Eq9}$$

where $\theta_h$ and $p_h$ are correlation parameters. Here, the correlation varies between 0 and 1 for the farthest and closest points, respectively. The aim in this optimization algorithm is to estimate the parameters $\mu$, $\sigma^2$, $\theta_h$ and $p_h$ for $h = 1,..,n$ that maximizes the likelihood function $L$:

$$L = \frac{1}{(2\pi\hat{\sigma}^2)^{k/2}|\mathbf{R}|^{1/2}} exp\left[-\frac{(\mathbf{y} - 1\hat{\mu})'\mathbf{R}^{-1}(\mathbf{y} - 1\hat{\mu})}{2\hat{\sigma}^2}\right] \tag{Eq10}$$

where $y$ is a vector of length $k$ with the values of the observed data at each of the sample points. By varying $\theta_h$ and $p_h$ to find their optimal values that maximizes the likelihood function $L$, we can calculate $\mu$ and $\sigma^2$ and, hence, can predict the value of $f$ at any point $x$.

## Pareto optimization

For multi-objective optimization goals, there may be a trade-off between different objectives. For example, increasing the dose of each antibiotic in a regimen to the maximal dose would result in a minimal sterilization time at the cost of a very high dose, which may lead to severe side effects. Similarly, a very low dose would minimize the total dose of a regimen; however, the granuloma would sterilize slowly, if at all. Both solutions are a part of a Pareto set, which contains (non-dominated) optimal solutions using *different* weights on the objectives. Therefore, we need to derive compromised solutions, deciding weights between the objectives within the algorithm (see Fig B in S1 Appendix for a detailed description of a Pareto set). By using the predicted objective functions, our algorithm selects a new regimen that maximizes the likelihood of expected improvement of the Pareto set. Specifically, the expected improvement criterion seeks the regimen(s) that maximize(s) the expected distance from the points currently in the Pareto front and lies in the blue shaded area in Fig B in S1 Appendix, where new solutions would dominate the current Pareto set [64,88].

## Optimization pipeline in *GranSim*

Our optimization pipeline started with exploring an initial set of 40 regimens for each set of 4-way combinations (HRZE, HRZM, RMZE, HMZE, HRME). We generated these 40 regimens using the LHS sampling scheme for the parameter space of doses for each individual antibiotic. These were varied between 0 to the double of the standard CDC dose [4], i.e., 10, 20, 50, 40 and 14 for H, R, Z, E and M, respectively. For each regimen, we simulated 30 granulomas (15 high-CFU and 15 low-CFU granulomas) each for 180 days (Table 1) and averaged their sterilization times to evaluate the objective function for each regimen. Then, by using the multi-objective surrogate-assisted optimization algorithm, we predicted the objective functions and one regimen that is expected to improve the current Pareto set. We simulated this new regimen using *GranSim* on the 30 total high- and low-CFU granulomas. This iterative process continued for 20 iterations, and one optimal regimen was simulated at the end of each iteration. At the end of this pipeline, we computed the Pareto front, i.e., the optimal non-dominating regimens.

## Nonhuman primate model for *in vivo* regimen experimental studies

Nine male Chinese cynomolgus macaques (*Macaca fascicularis*) (4–7 years of age) were dedicated to this study and were infected with virulent *M. tuberculosis* strain Erdman (8–21 CFU) via bronchoscopic instillation into a lower lobe. Three months post-infection, drug treatment was initiated and continued for 2 months, then the animals were necropsied. Animals were monitored daily for appetite and behavior and monthly for weight and erythrocyte sedimentation rate (a sign of inflammation). Gastric aspirate and BAL samples were cultured for Mtb to assess disease progression. An additional seven cynomolgus macaques (2 males, 3 females, 5–9 years of age, infected with Mtb Erdman) from a concurrent study were included here as historical untreated controls and necropsied 5 months post-infection.

Drug treatments were 1. isoniazid (H), rifampicin (R), pyrazinamide (Z) and ethambutol (E) (HRZE, N = 3); 2. isoniazid (H), moxifloxacin (M), pyrazinamide (Z) and ethambutol (E) (HMZE, N = 4); or 3. rifampicin (R), moxifloxacin (M), pyrazinamide (Z) and ethambutol (E) (RMZE, N = 2). Drug dosing as follows: H: 15 mg/kg; R: 20 mg/kg; Z: 150 mg/kg; E 55 mg/kg; M: 35 mg/kg. Drugs were provided daily in treats or by gavage. Macaques were treated for 2 months, and drugs were stopped one day before necropsy. See Table 1 for a list of treatment protocols.

$^{18}$F-fluorodeoxyglucose (FDG) PET-CT imaging was performed prior to treatment initiation and at 4- and 8-weeks post-treatment initiation. FDG is a glucose analogue, which is preferentially taken up by and retained in metabolically active cells and thus is useful as a proxy for inflammation. FDG uptake was quantified using the peak standard uptake value (SUV) associated with each granuloma, as previously described [52].

Detailed necropsies were performed using the final PET-CT scan as a map to isolate all lesions (granulomas, consolidations, etc.), uninvolved lung lobe samples, all thoracic lymph nodes, peripheral lymph nodes, spleen and liver. All samples were plated individually for Mtb on 7H11 plates, incubated for 3 weeks in a 5% CO2 incubator. Bacterial burden for each sample was calculated based on colonies counted on plates. Sum of all samples in thoracic cavity (lung, granulomas, lymph nodes) is reported as total thoracic CFU; total lung and total thoracic LN are also calculated.

## Supporting information

**S1 Appendix. Optimizing tuberculosis treatment efficacy: comparing the standard regimen with Moxifloxacin-containing regimens. Fig A. Visualization of simulated NHP granuloma FDG avidity using *GranSim*.** The structure indicates *FDG avidity* of a simulated granuloma using *GranSim*. The brightness scale of each microcompartment on the grid is calculated based on inflammation within *GranSim* measured here by TNF concentration levels and a weighted sum of each proinflammatory immune cell in the granuloma. See Methods for details. **Fig B. Schematic of Pareto Front Optimization.** The non-dominating solutions, i.e., the Pareto set (x's) for a problem with two objectives $f_1(x)$ and $f_2(x)$. The solid red line represents the Pareto front. Any solution that lies in the red shaded area would be a part of the Pareto set, whereas solutions in the blue shaded area would dominate the preexisting Pareto set and, hence, replace it [1]. **Fig C. Comparing 3-way simulated combinations of HRZEM using *GranSim*.** Comparison of simulations for (A-C) fraction of unsterilized granulomas and (D-F) sterilization times of 3-way combinations of HRZEM for (A and D) 100 high-CFU, (B and E) 100 low-CFU and (C and F) a combination of 100 high- and 100 low-CFU granulomas. (A-C) Moxifloxacin-containing regimens sterilize granulomas a lot faster initially than regimens without moxifloxacin in all cases, and this difference is more pronounced for (B) low-CFU granulomas. (D-F) Sterilization times shows a similar trend: granulomas treated with regimens containing moxifloxacin (red boxes) are cleared in a shorter time frames on average than regimens not containing moxifloxacin (black boxes). We performed significance tests between each possible pair of regimens with (red boxes) and without (black boxes) including moxifloxacin and show that regimens containing moxifloxacin are significantly more efficacious than regimens that do not include moxifloxacin (*p<0.0001, one-tailed paired t-test). The central red lines in box plots represent the median, whereas the bottom and the top edges of boxes represent 25$^{th}$ and 75$^{th}$ percentiles, respectively. **Fig D. Comparing 2-way simulated combinations of HRZEM using *GranSim*.** Comparison of simulations for (A-C) fraction of unsterilized granulomas and (D-F) sterilization times of 2-way combinations of HRZEM for (A and D) 100 high-CFU, (B and E) 100 low-CFU and (C and F) a combination of 100 high- and 100 low-CFU granulomas. (A-C) Moxifloxacin-containing regimens sterilize (B) low-CFU granulomas faster initially than regimens not containing moxifloxacin in all cases, but the same trend does not always hold for (A) high-CFU granulomas and (C) the combination of high- and low-CFU granulomas. (D-F) Sterilization times averaged over all granulomas for (D) high-CFU and (E) low-CFU granulomas, and (F) both groups combined (black boxes: regimens with no moxifloxacin, red boxes: regimens with moxifloxacin). We performed significance tests between each possible pair of regimens with (red boxes) and without (black boxes)

including moxifloxacin and showed that HM and RM are significantly more efficacious than regimens not including moxifloxacin (*p<0.005, one-tailed paired t-test). The central red lines in box plots represent the median, whereas the bottom and the top edges of boxes represent 25th and 75th percentiles, respectively. **Fig E. Data derived from NHP studies on total lung CFU**. The total CFU per NHP granulomas after NHP are treated with the corresponding regimens for two months demonstrate that most granulomas are sterilized by all regimens compared to the control (n = 107 granulomas from 7 animals in the control group, n = 117 granulomas from 4 animals in HMZE group, n = 52 granulomas from 3 animals in HRZE group, n = 34 granulomas from 2 animals in RMZE group). **Fig F. Simulation showing possibility of Paucibacilliary TB for 2 regimen treatments.** Total number of Mtb of high-CFU granulomas when treated with (A) ZM and (B) EM for 300 days. Treatment starts 300 days after the infection. Each line represents a granuloma simulation in *GranSim*. **Table A. Fractional inhibitory concentration (FIC) values of each regimen as predicted by INDI-GO-MTB [2].** These values are used in the drug combinations. **Table B. Predicted dosages for each antibiotic in the HRZM regimen that are optimal for minimizing dose and sterilization time (red dots in Fig 7A).** Red row indicates the regimen with CDC-recommended doses for each antibiotic (green dot in Fig 7A) [3]. The row labeled with a triangle indicates an optimal 3-way combination, where the optimal dose of Z is predicted as 0. **Table C. Predicted dosages of each antibiotic in HRME regimen that is optimal for dose and sterilization times (Red dots Fig 7B).** Red row indicates the regimen with CDC-recommended doses for each antibiotic (green dot in Fig 7B) [3]. The rows labeled with a triangle indicate an optimal 3-way combination, where the optimal dose of E is predicted as 0. **Table D. Predicted dosages of each antibiotic in HMZE regimen that are optimal for dose and sterilization times (Red dots Fig 7C).** Red row indicates the regimen with CDC-recommended doses for each antibiotic (green dot in Fig 7C) [3]. The row labeled with a triangle indicates an optimal 3-way combination, where the optimal dose of E is predicted as 0. **Table E. Predicted dosages of each antibiotic in RMZE regimen that are optimal for dose and sterilization times (Red dots Fig 7D).** Red row indicates the regimen with CDC-recommended doses for each antibiotic (green dot in Fig 7D) [3]. The rows labeled with a triangle indicate an optimal 3-way combination, where the optimal dose of E or Z is predicted as 0. **Table F. Predicted dosages of each antibiotic in HRZE regimen that are optimal for dose and sterilization times (Red dots Fig 7E).** Red row indicates the regimen with CDC-recommended doses for each antibiotic (green dot in Fig 7E) [3].
(DOCX)

**S1 Text. Changes to *GranSim* and PK/PD modeling that are included in this next-generation version. Fig A. *GranSim* Calibrated to non-human primate data from Flynn lab [5,6].** (A) T cell and (B) macrophage counts from a new set of granulomas sampled from calibrated parameter ranges span the range of *in vivo* data. Black dots are *in vivo* data from NHP granulomas, blue lines are the maximum, mean and minimum (from top to bottom) values of corresponding simulated cell counts from *GranSim* simulations, and the blue shaded area is between the minimum and maximum. **Fig B. Schematic representation of how antibiotics are partitioned within a grid microcompartment within *GranSim*.** Within a microcompartment, antibiotics can be caseum-bound ($DC_{caseum}$), be located within macrophages ($DC_{mac}$) or be free without binding to anything ($DC_{free}$), depending on the availability of caseous tissue and macrophages within that microgrid. ($DC_{caseum}$: drug concentration bound to caseum, $DC_{mac}$: drug concentration within a macrophage, $DC_{free}$: free drug concentration, green circle: macrophage, brown bleb: caseum, black circles: replicating extracellular Mtb, light green circle: intracellular Mtb, tan circle: nonreplicating Mtb). **Fig C. Calibration of *GranSim* PK/PD to**

**Moxifloxacin datasets.** Calibration of moxifloxacin (MXF) plasma and tissue PK to temporal (black dots in A-D) and spatial (see Fig 2B in [8]) data from human granulomas. Black lines in A-D are the maximum, mean and minimum (from top to bottom) values of MXF concentrations resulting from 100 *GranSim* simulations, and the black shaded area is between the minimum and maximum. Average simulated MXF concentrations in (A) blood, (B) granuloma, (C) uninvolved lung and (D) caseum agree with human data. (E) Spatial analysis of how MXF is distributed within a granuloma in *GranSim* indicates that MXF does not easily diffuse into caseum, which is consistent with MALDI-MS imaging of granulomas in [8]. All other drugs were calibrated using this same approach. **Table A. Pharmacodynamic (PD) parameters for each drug and Mtb type and sources for bactericidal assays used for calibration.** Emax values are reported as per *GranSim* timestep of 10 minutes.
(DOCX)

## Acknowledgments

We thank Dr. Simeone Marino for his contributions by developing the rules for FDG avidity measurement in *GranSim*. We thank Paul Wolberg for computational assistance and support. We are grateful to the veterinary and research technical staff in the Flynn lab for their work on this study. We thank Dr. Sriram Chandrasekaran for providing us with the FIC values for drug combinations. Simulations also use resources of the Advanced Research Computing (ARC), a division of Information and Technology Services (ITS) at the University of Michigan, Ann Arbor, the Extreme Science and Engineering Discovery Environment (XSEDE) supported by National Science Foundation Grant MCB140228 as well as the Expanse System at the San Diego Supercomputer Center.

## Author Contributions

**Conceptualization:** Maral Budak, Joseph M. Cicchese, Véronique Dartois, Jennifer J. Linderman, JoAnne L. Flynn, Denise E. Kirschner.

**Data curation:** Pauline Maiello, H. Jacob Borish, Alexander G. White, Harris B. Chishti, Jaime Tomko, L. James Frye, Daniel Fillmore, Kara Kracinovsky, Jennifer Sakal, Charles A. Scanga, Philana Ling Lin.

**Formal analysis:** Maral Budak, Joseph M. Cicchese.

**Funding acquisition:** Véronique Dartois, Jennifer J. Linderman, JoAnne L. Flynn, Denise E. Kirschner.

**Investigation:** Maral Budak, H. Jacob Borish, Alexander G. White, Harris B. Chishti, Jaime Tomko, L. James Frye, Daniel Fillmore, Kara Kracinovsky, Jennifer Sakal, Charles A. Scanga, Philana Ling Lin.

**Methodology:** Maral Budak, Denise E. Kirschner.

**Project administration:** Denise E. Kirschner.

**Resources:** Véronique Dartois, Jennifer J. Linderman, JoAnne L. Flynn, Denise E. Kirschner.

**Supervision:** Véronique Dartois, Jennifer J. Linderman, JoAnne L. Flynn, Denise E. Kirschner.

**Writing – original draft:** Maral Budak, Denise E. Kirschner.

**Writing – review & editing:** Maral Budak, Véronique Dartois, Jennifer J. Linderman, JoAnne L. Flynn, Denise E. Kirschner.

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
