## [Decision Letter · Decision Letter 0]

23 Mar 2023

Dear Ms Budak,

Thank you very much for submitting your manuscript "Optimizing tuberculosis treatment efficacy: comparing the standard regimen with Moxifloxacin-containing regimens" for consideration at PLOS Computational Biology. As with all papers reviewed by the journal, your manuscript was reviewed by members of the editorial board and by several independent reviewers. The reviewers appreciated the attention to an important topic. Based on the reviews, we are likely to accept this manuscript for publication, providing that you modify the manuscript according to the review recommendations.

Sincerely,

Adrianne L Jenner

Academic Editor

PLOS Computational Biology

Amber Smith

Section Editor

PLOS Computational Biology

Reviewer's Responses to Questions

**Comments to the Authors:**

Reviewer #1: This is a very well written manuscript that integrates computational and experimental approaches to advance the field of TB therapeutics, by addressing some of the hardest problems in the field, reducing treatment duration and optimizing dose. I'm attaching some questions below for the authors to provide written response back on and help improve the manuscript from the standpoint of the reader.

Line 169/170: Can the model be extended to predict relapse rate in patients after end of treatment ? Simulation results indicate one would expect a better outcome, but the translational results in the clinic indicate otherwise.

Line 243: What do you make of the differences between model and experimental results on metabolic effects of RMZE ?

Line 243: Are there markers of inflammation in the model and have these shown any results showing reduction in inflammation ?

Line 260: What other features of granulomas were varied in these simulations ? e.g. size, inflammatory, metabolic,…

Line 351: What PKPD information about each drug was used to do the dose predictions, and have you considered penetration depth ?

Discussion: Future work can consider testing a subset of the top functioning combinations in an animal model. Also consider expanding the model to include prediction of clinical relapse.

Reviewer #2: 1. Please clarify how did the authors derive adjusted concentration for 3 or 4 drug combinations?

2. With respect to the simulations with GranSim shown in Figure 6: As these were standard doses, and some of the combinations have been evaluated in literature in preclinical models, is there a similarity in trends for time to sterilization metric? Something like that could lend validity to the GranSim simulations.

3. Comparing GranSim simulations to NHP and humans (Figure 2,3): Does GranSim consider interspecies PK or PD parameter scaling, or differences in PD parameter within granulomas of the different species?

4. In Figure 1, Does each individual data point come from single NHP or multiple NHPs? Please elaborate?

5. With respect to Figure 5, what is the typical distribution of the ratio of high-low CFU granulomas observed clinically? Clinical relevance could add value.

6. When making the various comparisons, particularly in figures 2-6, provide a measure of how much change was observed along with a measure of variability and a statistic quantifying the differences.

**Have the authors made all data and (if applicable) computational code underlying the findings in their manuscript fully available?**

Reviewer #1: None

Reviewer #2: Yes

PLOS authors have the option to publish the peer review history of their article (what does this mean?). If published, this will include your full peer review and any attached files.

Reviewer #1: **Yes: **Karim Azer

Reviewer #2: No

Figure Files:

Data Requirements:

Reproducibility:

References:

---

## [Decision Letter · Decision Letter 1]

17 May 2023

Dear Ms Budak,

We are pleased to inform you that your manuscript 'Optimizing tuberculosis treatment efficacy: comparing the standard regimen with Moxifloxacin-containing regimens' has been provisionally accepted for publication in PLOS Computational Biology.

Best regards,

Adrianne L Jenner

Academic Editor

PLOS Computational Biology

Amber Smith

Section Editor

PLOS Computational Biology

Reviewer's Responses to Questions

**Comments to the Authors:**

Reviewer #1: The authors have successfully addressed all comments and updated the manuscript as necessary. The manuscript represents strong computational science and will help advance the field of TB drug combination design and translation into the clinic.

Reviewer #2: The authors have addressed all my comments and questions adequately.

**Have the authors made all data and (if applicable) computational code underlying the findings in their manuscript fully available?**

Reviewer #1: Yes

Reviewer #2: Yes

PLOS authors have the option to publish the peer review history of their article (what does this mean?). If published, this will include your full peer review and any attached files.

Reviewer #1: **Yes: **Karim Azer

Reviewer #2: No

---

## [Editor Report · Acceptance letter]

7 Jun 2023

PCOMPBIOL-D-22-01842R1 

Optimizing tuberculosis treatment efficacy: comparing the standard regimen with Moxifloxacin-containing regimens

Dear Dr Kirschner,

I am pleased to inform you that your manuscript has been formally accepted for publication in PLOS Computational Biology. Your manuscript is now with our production department and you will be notified of the publication date in due course.

With kind regards,

Zsofi Zombor
